# Chronic Administration of Melatonin: Physiological and Clinical Considerations

Donald Givler [1], Amy Givler [1], Patrick M. Luther [2], Danielle M. Wenger [3], Shahab Ahmadzadeh [4], Sahar Shekoohi [4,*], Amber N. Edinoff [5], Bradley K. Dorius [4], Carlo Jean Baptiste [4], Elyse M. Cornett [4], Adam M. Kaye [6] and Alan D. Kaye [4]

1 Department of Family Medicine, Louisiana State University Health Sciences Center at Shreveport/Monroe, Monroe, LA 71210, USA
2 School of Medicine, Louisiana State University Health Sciences Center at Shreveport, 1501 Kings Highway, Shreveport, LA 71103, USA
3 College of Medicine—Phoenix, University of Arizona, 475 N 5th St., Phoenix, AZ 85004, USA
4 Department of Anesthesiology, Louisiana State University Health Sciences Center at Shreveport, 1501 Kings Highway, Shreveport, LA 71103, USA
5 Harvard Medical School, Massachusetts General Hospital, Boston, MA 02114, USA
6 Department of Pharmacy Practice, Thomas J. Long School of Pharmacy and Health Sciences, University of the Pacific, Stockton, CA 95211, USA
* Correspondence: sahar.shekoohi@lsuhs.edu

**Abstract:** Background: Exogenous melatonin is commonly used to treat insomnia, other sleep problems, and numerous medical illnesses, including Alzheimer's disease, autism spectrum disorder, and mild cognitive impairment in adults and children. There is evolving information regarding issues with the use of chronic melatonin. Methods: The present investigation was a narrative review. Results: Melatonin usage has risen dramatically in recent years. Many countries only allow melatonin prescriptions. In the United States (U.S.), it is classified as a dietary supplement accessible over the counter and can be derived from animals, microorganisms, or, most commonly, made synthetically. No regulatory agency oversees its manufacturing or sale in the U.S. melatonin concentration of marketed preparations varies widely between product labels and manufacturers. Melatonin's ability to induce sleep is detectable. However, it is modest for most people. Sleep length appears to be less important in sustained-release preparations. The optimal dosage is unknown, and routinely used amounts vary substantially. Melatonin's short-term negative effects are minimal, resolve at medicine cessation, and do not usually prevent usage overall. Much research on long-term melatonin administration has found no difference between exogenous melatonin and placebo in terms of long-term negative effects. Conclusion: Melatonin at low to moderate dosages (approximately 5–6 mg daily or less) appears safe. Long-term usage appears to benefit certain patient populations, such as those with autism spectrum disorder. Studies investigating potential benefits in reducing cognitive decline and increased longevity are ongoing. However, it is widely agreed that the long-term effects of taking exogenous melatonin have been insufficiently studied and warrant additional investigation.

**Keywords:** melatonin; insomnia; sleep; aging; dietary supplement; autistic spectrum disorder

## 1. Introduction

Exogenous oral melatonin is used worldwide to treat insomnia and other sleep-related conditions. It is also commonly used to treat conditions such as autistic spectrum disorder (ASD), mild cognitive impairment (MCI), and Alzheimer's disease (AD). In recent years, melatonin use has increased significantly in both adults and children [1,2]. It is estimated that in the United States (U.S.), as many as 2.1% of adults and 6% of children, i.e., over 5 million adults and 4 million children, take melatonin at least monthly. Melatonin is available only by prescription in the United Kingdom, European Union, Japan, Australia.

However, it is available to purchase over the counter in the U.S. It is considered a dietary supplement by the Food and Drug Administration (FDA) in the U.S., and therefore, it Is not subject to the same regulations required of prescription pharmaceuticals. Subjugation to the investigation of dietary supplements by the FDA usually occurs on an incidence-based interval. Melatonin taken in low-to-moderate doses (e.g., 5 mg daily or less) for short periods appears safe and well-tolerated. However, concerns have been raised about the possible long-term effects of taking melatonin. Therefore, the present investigation focused on the effects of melatonin administration in both children and adults.

## 2. Literature Search

The present investigation was a narrative review. In 2022, we performed a comprehensive search for English-language studies related to the chronic administration of melatonin. Our investigation searched the following databases: Pub Med and Google Scholar. The study used the following combinations of keywords: melatonin, insomnia, sleep, aging, dietary supplement, and autistic/autism spectrum disorder. The investigation attempted to use as many recent manuscripts as possible—those within the last five years—but also included papers older than five years if they were particularly relevant to our topic. We also attempted to search for, use, and cite primary manuscripts whenever possible. A summary of the studies reviewed in this paper is outlined in Table 1.

**Table 1.** Characteristics of Cited Studies.

| Author (Year) (Citation) | Type of Study | Results and Findings | Conclusions |
|---|---|---|---|
| Kimland (2021) [1] | Study of the prevalence and incidence of melatonin prescription and long-term use in children and adolescents aged 0–17 in Sweden during 2006–2017. | In 2017, nearly 2% of the pediatric population in Sweden was dispensed at least 1 prescription of melatonin, representing a 15-fold increase for girls and a 20-fold increase for boys; nearly 80% had concomitant prescriptions of psychotropic medications. | The increase in melatonin use in children, often concomitant with psychotropic medications, suggests the need for further studies of the safety of long-term melatonin use. |
| Li (2022) [2] | Research letter regarding the use of melatonin in the U.S. from 1999–2018. | | Among U.S. adults, melatonin use increased from 0.4% to 2.1% between 1999 and 2018; use of melatonin in doses of greater than 5 mg per day also increased during the same period. |
| Tordiman (2017) [3] | Review of the effects of melatonin pharmacologically, physiologically, and pathologically. | | |
| Hardeland (2012) [4] | Review of melatonin levels during aging and in various diseases. | | Melatonin levels decrease during aging; reduced melatonin levels are also found in dementia, mood disorders, severe pain, cancer, and type 2 diabetes mellitus. |
| Gandhi (2015) [5] | Zebrafish (a diurnal vertebrate) lacking melatonin were studied to determine the effect on sleep. | Endogenous melatonin plays a significant role in promoting the initiation and maintenance of sleep in zebrafish. | Melatonin is required for the circadian regulation of sleep in a diurnal vertebrate (zebrafish). |
| Carrillo-Vico (2013) [6] | Review of the effects of melatonin in the immune system. | | Melatonin acts as an immune buffer. Acts as a stimulant under Basal or immunosuppressive conditions. Acts as an anti-inflammatory in exacerbated immune responses. |
| Kohlmeier (2015) [7] | Description of the metabolism of melatonin. | | |

**Table 1.** *Cont.*

| Author (Year) (Citation) | Type of Study | Results and Findings | Conclusions |
|---|---|---|---|
| Ma (2004) [8] | Study aimed to unveil the finer details of melatonin metabolism. | Melatonin was found to be metabolized by CYP1A2 principally, and for the first time found that CYP1B1 plays a part in metabolism. | Presence of CYP1B1 is extrahepatic meaning that melatonin is metabolized both in the large intestine and the brain. |
| NCCIH (2022) [9] | General overview on melatonin. Seemingly meant for patients. | | |
| Erland (2017) [10] | Review of frequency of melatonin use in U.S. children and adults and the variability of the actual melatonin content of available supplements. | | The authors recommend that melatonin be regarded as a medicine in the U.S., available only by prescription and that the content and purity of melatonin supplements be more closely monitored. |
| Grigg-Damberger (2017) [11] | 30 commercial melatonin supplements were quantified for melatonin and serotonin. | Melatonin content was found to range from −83% to +478% of the labeled content; lot-to-lot variability of the same product varied by as much as 465%; serotonin was found in 26% of supplements tested. | Label claim and actual melatonin content of available melatonin products vary widely, and contamination with serotonin is common. |
| Jan (2008) [12] | Letter to the Editor regarding the status of melatonin as a "supplement" in the U.S. | | The writers recommend that melatonin should be regarded as a medication in the U.S., made available only by prescription, and regulated the same way as other pharmaceutical products. |
| Buscemi (2005) [13] | Meta-analysis of the efficacy and safety of exogenous melatonin for primary sleep disorders. | Use of melatonin reduced average sleep onset latency in all patients by an average of 11.7 min and by 38.8 min in patients with delayed sleep phase syndrome. | Melatonin reduces sleep onset latency by a small amount in most patients. |
| Rossignol (2014) [14] | Review of the use of melatonin in individuals with autistic spectrum disorder (ASD). | | Melatonin use in ASD can safely decrease sleep onset latency, increase total sleep duration, and improve daytime behaviors. |
| Rossignol (2011) [15] | Review and meta-analysis of the use of melatonin in autistic spectrum disorders (ASD). | | Melatonin administration in ASD is associated with improved sleep parameters, better daytime behavior, and minimal side effects. |
| Zambreli (2021) [16] | Review of the effects of antioxidant agents, including melatonin, on sleep in autistic spectrum disorder (ASD). | Treatment with immediate-release melatonin at doses of 2–10 mg/day was effective in shortening sleep-onset latency (SOL), reducing the number of awakenings per night and bedtime resistance, and increasing total sleep time; 2–10 mg/day of prolonged-release melatonin had similar effects on sleep and decreased disruptive behaviors and parenting stress. | Immediate-release and controlled-release melatonin are safe and effective in improving sleep in children with ASD; controlled-release melatonin decreases disruptive behavior and parenting stress. |
| Attia (2020) [17] | Case-control study of 25 boys aged 14–18 years old with constitutional delay of puberty (CDP) and 25 age-matched individuals to study melatonin levels. | Melatonin is significantly higher in boys with CDP compared to age-matched boys, and melatonin levels are negatively correlated with testosterone and FSH. | Melatonin is associated with the delayed onset of puberty onset and may be a cause of the delay. More research is needed to assess its role in hormonal development and puberty onset. |
| Blaise (2014) [18] | Review of pharmacological and nonpharmacological management of sundowning in patients with AD. | | The number of studies supporting treatment of sundowning in AD patients is sparce and in need of further study. |

**Table 1.** *Cont.*

| Author (Year) (Citation) | Type of Study | Results and Findings | Conclusions |
|---|---|---|---|
| Cardinali (2002) [19] | Study on 45 AD patients suffering from sleep disturbances, administered 6 mg melatonin for 4 months. | Improved sleep and decreased sundowning. | Melatonin seems to be effective in ameliorating sleep disturbances and sundowning. |
| Lee (2019) [20] | Review of the neuroprotective effects of melatonin in cerebral ischemia, Alzheimer's disease (AD), and depression | | Exogenous melatonin prolongs sleep time in patients with AD and dementia; anti-depressant effects of melatonin have not been established; the role of melatonin in the treatment of cerebral ischemia requires further study |
| Cardinali (2019) [21] | Review of the use of melatonin in the management of Alzheimer's disease (AD) and Parkinson's disease (PD). | Melatonin improved the quality of sleep, decreased "sundowning", and improved cognitive performance in AD patients, and it improved the quality of sleep in PD patients with no effect on motor symptoms. | Melatonin improves the quality of sleep in AD and PD patients and is useful for the treatment of insomnia in these patients; its effect on cognitive performance in AD requires further study. |
| Cardinali (2012) [22] | Retrospective study of 96 patient with mild cognitive impairment (MCI) treated with 3–9 mg of immediate-release melatonin at bedtime for up to 3 years. | MCI patients treated with melatonin scored better on multiple neuropsychological assessments and were treated with benzodiazepines less frequently. | Melatonin can be a useful medication as an adjunct to standard medications in the treatment of MCI. |
| Medeiros (2007) [23] | A randomized, double-blind, placebo-controlled study of 18 patients with Parkinson's disease given 3 mg of melatonin 1 h before bedtime for 4 weeks. | Melatonin significantly improved subjective quality of sleep, but not objective quality as evaluated by polysomnography; melatonin did not improve motor dysfunction. | Melatonin improves subjective quality of sleep in PD patients. |
| Iyer (2020) [24] | A randomized, double-blind, placebo-controlled trial of 3 mg or 10 mg in 62 children with persistent post-concussion syndrome (PPCS) over 28 days. | Melatonin treatment did not result in overall recovery from PCCS; however, it did improve subjective and objective sleep parameters. | Melatonin improves sleep parameters in PPCS patients; the long-term benefits of melatonin treatment on overall PPCS recovery needs further study. |
| Zwart (2018) [25] | A follow-up study of 69 children aged 6–12 years with chronic sleep onset insomnia (CSOI); the overall average treatment duration was 7.1 years; sleep timing, sleep quality, adverse events, and reasons for cessation of therapy were assessed. | Long-term melatonin therapy appears to be safe after an average of 7.1 years; adverse effects were few and mild; there was no effect on long-term sleep quality; there were subjective concerns about delay in onset of puberty in 31.3% of those studied (compared to 17% in the general population). | Long-term melatonin therapy appears to be safe; there is subjective concern about possible delay in onset of puberty (this requires further study). |
| Hoebart (2009) [26] | Parents of 105 children with attention-deficit/hyperactivity disorder (ADHD) and chronic sleep onset insomnia (CSOI) who had been treated with melatonin over a mean 3.7-year period responded to a structured questionnaire with a 93% response rate. | No serious adverse effects or treatment-related co-morbidities were reported; long-term melatonin treatment was judged to be effective against sleep problems in 88% of cases; there were improvements in behavior and mood in 71% and 61%, respectively. | Melatonin is an effective and safe therapy for the long-term treatment of CSOI in children with ADHD. |
| Carr (2007) [27] | Placebo-controlled, double-blind, cross-over trial of controlled-release melatonin in 44 children with neurodevelopmental disabilities and treatment-resistant circadian rhythm sleep disorders for up to 3.8 years. | Subjective improvement in sleep, behavior and learning were reported by caregivers; adverse reaction to melatonin therapy and development of tolerance were not evident. | Controlled-release melatonin is safe and effective for children with neuro-developmental disabilities and sleep maintenance difficulties. |

**Table 1.** *Cont.*

| Author (Year) (Citation) | Type of Study | Results and Findings | Conclusions |
|---|---|---|---|
| Wasdell (2008) [28] | Randomized, placebo-controlled crossover trial of 51 children aged 2–18 years with neurodevelopmental disabilities (NDD), including autistic spectrum disorder (ASD) and with delayed sleep phase syndrome (DSPS) and impaired sleep maintenance (ISM); children were treated with 5 mg of controlled-release melatonin, increased as needed for optimal beneficial effects. | Those receiving melatonin had improvement in total nighttime sleep and sleep latency of approximately 30 min in 47 of 61 children; there was also a reduction in family stress; there was no evidence of significant side effects. | Controlled-release melatonin is safe and effective in improving both sleep latency and sleep duration in children with NDD, including ASD and with DSPS and ISM. |
| Maras (2018) [29] | A double-blind, randomized placebo-controlled study of 19 children aged 2–17.5 years with autistic spectrum disorder (ASD) and neurogenetic disorders (NGD) given 2–10 mg of prolonged-release melatonin for 52 weeks. | In those given melatonin, sleep latency, total sleep time, and sleep quality were improved; 5.3% of subjects reported fatigue, and 3.2% reported mood swings. | Prolonged-release melatonin is safe and efficacious for long-term treatment (up to 52 weeks) of insomnia in children with ASD and NGDs. |
| Gringras (2017) [30] | Randomized, double-blind, placebo-controlled study of 125 children and adolescents with autistic spectrum disorder (ASD) and neurogenetic disorders (NGD) treated with 2–5 mg of prolonged-release melatonin at bedtime for 13 weeks. | In those patients given melatonin, sleep latency decreased by 39.6 min, and length of sleep increased by 57.5 min compared to placebo; somnolence, headache, and fatigue were reported as side effects; no tolerance was observed. | Prolonged-release melatonin decreases sleep latency and increases the length of sleep in children and adolescents with ASD and NGD with only mild side effects. |
| Van Geijlswijk (2011) [31] | A follow-up research study of children aged 6–12 years with chronic idiopathic childhood sleep onset insomnia (CSOI) treated for an average of 3.1 years with 0.3–10 mg of melatonin. | There was no significant difference in puberty development, social development, and mental health scores as compared with the general population; adverse effects occurred infrequently and led to cessation of melatonin use in 1.6%. | Melatonin treatment in children with CSOI can be sustained over an average of 3 years without significant adverse effects. |
| Seabra (2000) [32] | Randomized, double-blind, placebo-controlled study of the toxicology of chronic melatonin treatment in 40 male adults aged 22–55 years administered 10 mg of melatonin for 28 days. | Analysis of multiple parameters (polysomnographic recording, somnolence scale, sleep diary, and clinical laboratory examinations) showed no difference between placebo and melatonin groups. | Melatonin appears to be safe when administered in a dose of 10 mg daily for 28 days. |
| Andersen (2016) [33] | Review of the safety of exogenous melatonin in humans. | | The use of melatonin in non-pregnant adults is safe; adverse effects are mild; long-term safety in children and adolescents should be studied further. |
| Van der Heijden (2007) [34] | Randomized, double-blind, placebo-controlled study of 105 children aged 6–12 years with attention-deficit/hyperactivity disorder (ADHD) and chronic sleep-onset insomnia (SOI) given 3 or 6 mg of melatonin for 4 weeks. | In melatonin-treated children, sleep onset advanced by an average of 26 min, and total time asleep increased by 19.8 min as compared to placebo; there were no significant adverse events and no significant effect on behavior, cognition, or quality of life. | Melatonin induces clinically relevant advances of sleep onset and increased total sleep time in children with ADHD and SOI with no apparent negative effects. |
| Wirtz (2008) [35] | Randomized, single-blinded, placebo-controlled study of 46 healthy men aged 20–34 years old treated with 3 mg oral melatonin or placebo pill to assess effect on plasma levels of procoagulants. | Subjects in the melatonin administration group had significantly lower levels of factor VIII and fibrinogen compared to the placebo group. | The use of melatonin is associated with a significant decrease in certain procoagulant factors (factor VIII, fibrinogen) in a likely dose-response relationship; melatonin should be studied as a potential therapeutic agent for patients at risk of ASCVD. |

**Table 1.** *Cont.*

| Author (Year) (Citation) | Type of Study | Results and Findings | Conclusions |
|---|---|---|---|
| Ashy (2022) [36] | Review Summarizing the consequences of COVID-19 infection on activity and function of coagulation. The effect of melatonin in this process. | | Melatonin may help protect against virally induced coagulopathies in COVID-19 patients. |
| Otmani (2008) [37] | Randomized, double-blind, placebo-controlled, and 4-way crossover study of 16 patients aged 55 years and older treated with 2 mg prolonged-release melatonin, 10 mg zolpidem, and the 2 combined; psychomotor functions, memory recall, and driving skills were evaluated at 1 and 4 h post-dosing. | Prolonged-release melatonin alone did not impair psychomotor functions, memory recall, or driving skills; zolpidem did impair all three parameters, impairments that were exacerbated with co-administration of melatonin. | Prolonged-release melatonin alone does not significantly impair psychomotor functions, memory recall, or driving skills; melatonin should be used cautiously in combination with zolpidem. |
| Tarocco (2019) [38] | Review of the effects of melatonin and its potential role and clinical implications for newborn care and pathologic conditions. | Repeated studies demonstrate that melatonin use is associated with a decrease in oxidative stress; melatonin is not solely made in the pineal gland but also in other tissue types and organ systems like the retina, GI tract, and immune system. | Melatonin may have some important anti-oxidative effects, such as reducing inflammatory biomarkers, providing neuroprotection, which may have a role in improving clinical conditions in neonates. |
| Gimenez (2022) [39] | Review of the effects of melatonin on the aging process and its associated diseases, such as cardiovascular and neurodegenerative diseases. | Melatonin can decrease mitochondrial dysfunction and cellular aging by modulating the sirtuin1 pathway, limiting the oxidation of cardiolipin, upregulating Nrf2 and downregulating NF-kB, and suppressing proinflammatory markers such as NO, COX-2, NLRP3, and B-amyloids. | The pharmacokinetic features of melatonin that make it anti-oxidative and anti-inflammatory suggests that melatonin may be considered for its therapeutic use as an anti-aging agent. |
| National Institute of Aging (2023) [40] | "How Biomarkers Help Diagnose Dementia" is an article published by the NIH that details the biomarkers and biomarker tests available and studied for dementia research purposes. | Alzheimer's disease is diagnosed and monitored by tests including CT, MRI, and PET scans looking for the presence of amyloid beta plaques, tau fibers, and fluorodeoxyglucose. | Several types of brain scans and biomarkers exist to help diagnose Alzheimer's disease and other related dementias; the discovery of new biomarkers is allowing researchers to make advancements in the field of dementia. |
| Nous (2021) [41] | A systematic review of 20 studies analyzing blood and CSF melatonin levels in patients with Alzheimer's disease (AD) compared to healthy controls. | A significant reduction in CSF levels, nocturnal blood, and nocturnal saliva levels of melatonin were found in patients with AD compared to controls; pineal gland and ventricular CSF melatonin levels have a strong correlation, while the relationship between blood and CSF melatonin levels needs further investigation. | There is indeed altered melatonin production, particularly decreased melatonin blood and CSF levels, as we age, and this reduction may become more significant in patients with AD; altered melatonin production in AD may help to explain the biological basis for circadian rhythm disturbances and sundowning effect commonly seen in patients with AD. |
| Alz.org (2023) [42] | "Mild Cognitive Impairment (MCI)" is an article published by the Alzheimer's Association that describes the wide spectrum of cognitive changes and medical workup for MCI. | MCI is a clinical diagnosis, and when individuals with MCI have a PET scan or CSF test that detects amyloid beta protein, these individuals are considered to have MCI secondary to Alzheimer's disease; aducanumab and lecanemab are newly approved by U.S. FDA for the treatment of early Alzheimer's disease. | MCI causes cognitive changes that may or may not be an early sign of Alzheimer's disease. A medical workup of MCI is critical to attempt to determine the root cause of MCI, e.g., a new medication or an irreversible neurodegenerative disease like Alzheimer's disease. |

**Table 1.** *Cont.*

| Author (Year) (Citation) | Type of Study | Results and Findings | Conclusions |
|---|---|---|---|
| Sumsuzzman (2021) [43] | A systemic review and meta-analysis of 22 studies investigating the impact of melatonin on cognition from randomized-control trials of oral melatonin treatment for Alzheimer's disease, insomnia, and/or healthy subjects. | Melatonin use among patients with mild-stage Alzheimer's disease significantly improved cognition (measured by MMSE score); chronic nighttime melatonin use among healthy subjects improved memory without any negative cognitive effects and may potentially improve cognition. | Long-term melatonin use is associated with positive cognitive outcomes and may be of particular benefit in patients with mild-stage Alzheimer's disease. |
| Boafo (2019) [44] | Review of effect of long-term use of melatonin in pre-pubertal children on the timing of puberty. | | The effect of long-term use of melatonin of pubertal timing is understudied. |
| Sheldon (1998) [45] | Research letter reporting on a study of 6 children aged 9 months to 18 years with multiple neurological deficits and chronic severe sleep complaints treated with 5 mg of melatonin. | Melatonin improved sleep-onset latency, sleep continuity, and total sleep time in five of six patients; however, the study was suspended because four of six patients had increased seizure activity. | Melatonin may increase seizure activity in children with multiple neurological deficits and should be used with caution. |
| Peled (2012) [46] | Review of melatonin utility in 6 children aged 2–15 years old with severe, intractable seizures. | Melatonin improved seizure activity in five of six children with intractable seizures. | Melatonin may be an effective adjunct to standard anti-epileptic therapy. |
| Maghbooli (2022) [47] | A cross-sectional, randomized, double-blind, placebo-controlled clinical trial of add-on melatonin therapy for 60 patients with epilepsy with idiopathic generalized tonic-clonic seizures alone (EGTCS). | The addition of melatonin to valproic acid treatment led to a significant decrease in mean severity score of epilepsy and improved sleep quality; the number of attacks and EEG results did not significantly change with melatonin vs. placebo. | Melatonin may be a useful adjunctive therapeutic agent for patients with EGTCS, given its ability to reduce severity of epilepsy and enhance sleep quality. |
| Frisher (2016) [48] | A retrospective cohort study of 1,377 patients ages 45 and older with at least 3 melatonin prescriptions over a 2-year period; records were reviewed for fracture risk. | Prescribed melatonin was associated with a significantly increased risk of fracture after adjusting for potential confounders. | Melatonin use may increase the risk of fractures in older adults. |
| Lui (2018) [49] | A randomized, double-blind, placebo-controlled observation study to determine the effect of melatonin on postural control and cognitive performance in 34 adults aged 60–71; testing while dual-tasking (concurrent physical and cognitive tasks) was performed before and after a 3 mg dose of immediate-release melatonin. | There was a significant decrease in postural control after taking melatonin but no change in cognitive performance. | A single dose of melatonin may disturb postural control in older adults; precautions may be necessary to decrease the risk of falls in older adults after taking melatonin. |
| Gooneratne (2012) [50] | A randomized, double-blind, placebo-controlled study of pharmacokinetics of low (<0.5 mg) and higher dose (>2 mg) combined immediate- and controlled-release melatonin in 27 adults over 65 years of age. | In those given higher dose melatonin, melatonin levels remained elevated for an average of 10 h. | Melatonin levels in older adults given higher dose (>2 mg) combined immediate and controlled release melatonin may remain elevated beyond the typical sleep period. |
| Whittom (2010) [51] | 8 adults aged 38–63 years with restless legs syndrome (RLS) received exogenous melatonin or bright light exposure; the effect on motor and sensory manifestations was measured. | Melatonin administration resulted in an increase in motor manifestations of RLS; bright light exposure resulted in a small but significant decrease in sensory symptoms. | Melatonin causes an increase in motor manifestations of RLS. |

**Table 1.** *Cont.*

| Author (Year) (Citation) | Type of Study | Results and Findings | Conclusions |
|---|---|---|---|
| Aurora (2012) [52] | An evidence-based systematic review and meta-analysis of treatment of restless legs syndrome and periodic limb movement disorder (PLMD) in adults. | 3 mg of melatonin decreased leg movements, sleep arousal, and subjective well-being in PLMD. | Melatonin is useful in the treatment of periodic leg movement disorder (PLMD) in adults. |
| Campos (2004) [53] | A randomized, double-blind, placebo-controlled study of the use of melatonin in 22 adults with mild to moderate asthma. | Melatonin improved subjective sleep quality with no effect on asthma symptoms, use of relief medication, or daily peak expiratory flow. | Melatonin can safely be used in adults with mild to moderate asthma. |
| Carman (1976) [54] | Double blind cross-over study of 6 moderately to severely depressed patients and 2 patients with Huntington's chorea were treated with PO or IV melatonin at doses of 150–1600 mg daily. | All patients studied showed an increase in psychological symptoms (depression, psychosis, anger, and anxiety) during treatment with melatonin. | Melatonin may worsen depression, psychosis, anger, and anxiety in moderately to severely depressed patients at higher doses than are usually used for the treatment of sleep disorders. |
| Hansen (2014) [55] | Review, qualitative synthesis, and meta-analysis of the therapeutic or prophylactic effect of exogenous melatonin on depression and depressive symptoms. | | There is no clear evidence that melatonin worsens or improves depression or depressive symptoms. |

## 3. Endogenous Melatonin Physiology

The mammalian pineal gland produces melatonin (N-acetyl-5 methoxytryptamine) under direct innervation from the suprachiasmatic nucleus of the hypothalamus (the central circadian pacemaker). Plasma melatonin concentrations are typically low during the day, rising approximately 2 h before bedtime, remaining elevated through the normal sleep period, and declining rapidly approximately 1 h before wake time [3]. Melatonin levels drop with age, and reduced melatonin levels are found in individuals with mood disorders, dementia, severe pain, cancer, type 2 diabetes mellitus, and autistic spectrum disorder [4].

The role of endogenous melatonin in normal human sleep patterns is not well understood; however, it is generally believed to promote the initiation and maintenance of sleep at night in diurnal mammals [5]. Low nocturnal melatonin levels in humans are often associated with sleep difficulties. Interestingly, in nocturnal mammals, melatonin is still produced at night when they are most active. The significance of this observation is not clear. Melatonin is additionally involved in thymus regulation and immunocompetence. In a larger sense, breakdown of the immune system in the setting of decline in melatonin can result in immunosenescence and consequent inflammation and decline of the brain and body with clinical evidence of anxiety, depression, cognitive decline, pain, and fatigue. These processes can be seen as early inflammatory consequences of degeneration that ultimately result in AD and reduced longevity [6].

## 4. Pharmacokinetics of Melatonin

After intravenous administration, melatonin will reach peak plasma levels in approximately 0.5 to 0.6 min. Oral administration results in peak plasma concentration in approximately 60 min. Bioavailability of oral melatonin ranges from 10 to 56% with a mean of 33% [3]. Given the unregulated nature of melatonin, there may be significant variation of bioavailability depending on the source of the exogenous melatonin and the metabolic rate of the patient of which it is administered.

Melatonin is hydroxylation to 6-hydroxymelatonin carried out by CYP1A2 and some other related cytochromes within the liver [7]. There are also reports of metabolism being carried out by CYP1B1, an extrahepatic cytochrome that is capable of the same hydroxylation carried out by CYP1A2 [8]. After being hydroxylated, the newly formed

6-hydroxymelatonin will then be conjugated mostly to sulfuric acid, a small percentage will either be conjugated to glucuronic acid, and then excreted in [3].

## 5. Uses of Exogenous Melatonin

Oral melatonin supplementation has been advocated for treating both short-term and chronic insomnia. This includes both circadian phase delay (delayed sleep-wake phase disorder) and sleep-onset insomnia in adults and children. It is also commonly used for other conditions, including non-24-h sleep-wake rhythm disorder, sleep-wake disturbances in shift workers, jet lag after eastward travel, and sleep onset and maintenance in children and adolescents with autistic spectrum neurogenetic disorders, as well as those with attention-deficit or attention-deficit hyperactivity disorder [9]. While there are many possibly effective uses of melatonin, each indication needs repeated studies in order to confirm its therapeutic value in said area.

## 6. Available Formulations of Melatonin

Melatonin is available in immediate-release and sustained-release oral tablets, liquid, intranasal spray, transdermal, and sublingual formulations. Recommended doses vary widely, from as low as 0.1 mg to greater than 20 mg. The precise optimal dose for the various melatonin uses is unknown, although typical doses are in the 1–5 mg range. It has been suggested that doses below 1 mg may be as effective as higher amounts. Because the FDA does not regulate melatonin, commercially available formulations can vary in accuracy and may even contain porcine. Individuals may reject these porcine-containing melatonin formulations due to religious or dietary purposes. One study found that actual melatonin content varied from 83% to 478% of the labeled content, with significant lot-to-lot variability, and 26% of the samples contained serotonin as a contaminant [10]. Some writers have recommended that melatonin be more closely regulated in the U.S. and its use restricted to prescription-only [11,12]. A melatonin-receptor agonist, ramelteon (Rozerem), is available in the U.S. by prescription to treat insomnia.

## 7. Effectiveness of Exogenous Melatonin

The most common use for exogenous melatonin is insomnia. Melatonin is widely believed to have "soporific" (sleep-inducing) effects. Multiple studies have confirmed that an immediate-release formulation taken 30 min before the desired bedtime reduces the time to fall asleep by as little as one minute to over 30 min. [13]. This reduction in time to sleep onset is statistically significant but may or may not be clinically significant. Evidence of the ability of exogenous melatonin to increase clinically meaningful sleep duration is less consistent, although this is more likely to occur with a sustained-release formulation. In addition, it appears that some people benefit from the soporific effects of melatonin more than others for reasons that are not clear.

Other common uses of melatonin include treating children with neurodevelopmental disabilities, including ASD, and adults with MCI and AD. Multiple studies of melatonin (e.g., both immediate-release and extended-release) in ASD have shown that melatonin decreases the time to sleep onset, increases sleep duration, and decreases nighttime awakenings. The nighttime administration of melatonin was also shown to improve daytime behaviors in children with ASD [14–16]. However, some research has shown that melatonin use in children may affect hormonal development, mainly delaying the onset of puberty [17]. Despite the lack of research on the long-term effects of melatonin administration in children, Multiple studies of adults with AD have also shown improved sleep quality and decreased "sundowning" [18,19]. Some studies have shown an improvement in cognitive function in these patients, while others have been inconclusive [20–22]. In patients with Parkinson's disease, melatonin improved the quality of sleep as measured by subjective standards, although not by objective criteria (polysomnography), and it did not affect motor symptoms [23]. Melatonin has also been studied for its neuroprotective effects and how it affects neuroplasticity. Specific areas of clinical interest include cerebral

ischemia, neonatal hypoxic-ischemic encephalopathy [20], and children with persistent post-concussive symptoms [24]. Additional studies are needed to verify the effectiveness of melatonin for these indications and determine optimal dosing.

## 8. Short-Term Side Effects of Taking Melatonin

Exogenous melatonin appears to be very safe in moderate doses (less than 5 mg) for short periods [14,25–28]. All commonly reported side effects are minor and resolve once medication usage is ceased [29–31]. The most frequent side effects are headache and somnolence (fatigue). However, one study found no difference in the incidence of these symptoms between the placebo and melatonin groups [32]. Other reported side effects include hypotension, hypertension, dizziness, gastrointestinal upset, vivid dreams, irritability, bedwetting (in children), and depression [33]. However, a meta-analysis by Buscemi et al. [13] and a study by Van der Heijden et al. [34] found no significant differences in side effects between melatonin and placebo.

Multiple potential drug interactions have been reported. Tricyclic antidepressants, fluvoxamine, cimetidine, ciprofloxacin, caffeine, and oral contraceptives may decrease melatonin metabolism, and thus, increase serum melatonin concentrations. In contrast, carbamazepine, omeprazole, beta-blockers, nonsteroidal anti-inflammatory agents, alcohol, and smoking may decrease serum concentrations. Melatonin can increase the risk of bleeding in those taking exogenous doses as related to recent studies that show that there may be a dose-response relationship between coagulation activity and plasma levels of melatonin [35]. In this study, Wirtz et al. found decreased plasma levels of coagulation factors FVIII:C and Fibrinogen 1 h after oral administration of 3 mg melatonin [35]. There is little primary literature that targets melatonin and warfarin interaction. Of the few studies performed, there was a small cohort of 10 patients evaluated for potential melatonin-warfarin interaction at Massachusetts General Hospital in Boston. Noha Ashy and K.V. Stroff found increased measures of INR (international normalized ratio) and PT (prothrombin time) in patients that were administered both melatonin and warfarin [36]. Melatonin can also decrease the effectiveness of extended-release nifedipine for blood pressure control. It can potentially decrease the effectiveness of immunosuppressant medication used after organ transplants. Melatonin used with benzodiazepines or zolpidem may cause increased sedation [37].

Exogenous melatonin does not appear to reduce endogenous melatonin production via rebound insomnia. Withdrawal symptoms have not been reported with cessation of exogenous melatonin [26]. Safety during pregnancy and breastfeeding is unknown.

## 9. Melatonin: Usage and Considerations

### 9.1. Anti-Aging and Oxidation Protection

Mitochondrial dysfunction is related to several age-related diseases, such as AD and cardiovascular disease, caused by oxidative stress induced by the buildup of ROS. The production of melatonin is not only found within the pineal gland. Some studies have found melatonin production within bone marrow, retina, and gastrointestinal tract [38]. These extra pineal sources of melatonin are produced within mitochondria of the tissues above, and this production significantly decreases with age, with mitochondrial dysfunction increasing. A thorough review published in 2022 discussed the causative factors of aging and how the decrease in melatonin production may be an important factor in aging [39]. Gimenez et al. summarized how exogenous administration of melatonin reduces the oxidation of compounds such as cardiolipin, implicated in cardiovascular disease, and its inhibition in proinflammatory events such as COX-2, amyloid beta toxicity, and mTOR signaling (refer to Figure 1) [39]. The reduction of oxidation of compounds related to mitochondrial function may be implicated in an "anti-aging" effect. Essentially, melatonin may preserve mitochondrial function, thus preventing the vicious cycle of oxidative damage that arises in concordance with aging and age-related diseases that have exhibited an ever-increasing incidence within the U.S. population.

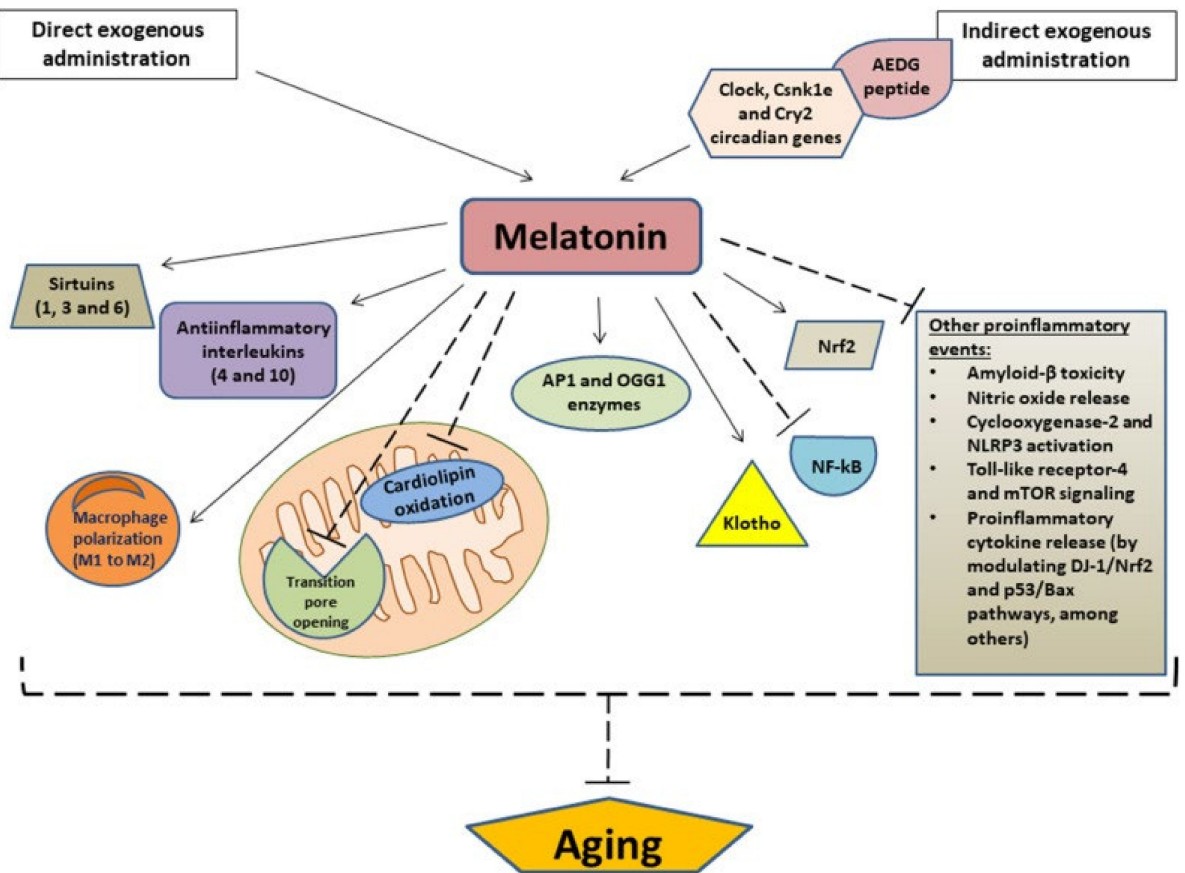

**Figure 1.** Gimenez et al. details the downstream and potential anti-aging effects of exogenous administration of melatonin. Arrows indicate stimulation, and dashed lines indicate inhibition. AEDG, (Ala-Glu-Asp-Gly) peptide; APE1, Apurinic/apyrimidinic endonuclease 1; OGG1, (8-Oxoguanine glycosylase); Nrf2, nuclear factor erythroid 2-related factor 2; NF-κB, nuclear transcription factor kappa B; NLRP3, NOD-like receptor family pyrin domain-containing 3; mTOR, mammalian target of rapamycin [31].

### 9.2. Melatonin as a Potential Marker in Alzheimer's Disease

The diagnosis and determination of progression of AD is obtained via many tests, including PET scanning for the uptake of flurodeoxyglucose indicating the presence of amyloid beta plaques and tau fibers [40]. Historically, it has been troublesome to use biomarkers within liquid media to track the progression of AD. A 2021 study found a significant difference in nighttime circulating levels of melatonin in AD and age-matched individuals, suggesting this may be a new biomarker to consider in clinical practice; however, more studies need to be performed in order to further evaluate the use of circulating melatonin as a diagnostic tool for this disease [41].

### 9.3. Therapeutic for Mild Cognitive Impairment

MCI is a condition that involves impairment of an individual's memory, reasoning, or perception. This condition is implicated in the progression of AD as an early-stage finding that can be diagnosed with concurrent imaging showing beta-amyloid plaques or tau fibers on a PET scan [42]. A systematic review and meta-analysis of over 22 randomized control trials found that melatonin usage significantly increased the Mini-Mental State Examination score in patients suffering from mild AD [43]. This finding suggests that melatonin may be an effective adjuvant therapeutic option for improving MCI in patients suffering from mild AD.

### 9.4. Delay of Puberty

There has been concern that long-term exogenous melatonin might delay children's sexual maturation. This stems from the observation that the highest nighttime melatonin

levels are found in very young individuals (1–3 years) and that levels drop progressively until adolescence. The drop in nocturnal serum melatonin levels parallels the sexual maturation process. Therefore, it has been hypothesized that artificially keeping melatonin levels high might not allow the triggering of important pubertal developmental steps.

Boafo et al. [44] analyzed three human studies that tracked pubertal timing along long-term exogenous melatonin use. The primary focus was melatonin effectiveness or dosing in all three studies, with pubertal timing as a secondary outcome. One study [31] showed no effect on pubertal development based on Tanner scores. A second study [25] showed "perceived pubertal timing as late" (a subjective delay) in melatonin users compared to controls. In a third study [27], out of 44 children with severe neurodevelopmental difficulties, 5 developed precocious puberty before starting melatonin therapy; however, there was no apparent impact on the onset of puberty in the remaining 39 subjects. Boafo et al. [44] concluded that the number of patients studied was too small to draw reliable conclusions, patient follow-up was incomplete, and measures of pubertal timing were poor. They concluded that the role of melatonin in sexual maturation and the timing of puberty is understudied, and thus, that additional research is necessary.

### 9.5. Effect on Seizures

There has been concern that melatonin may increase the incidence of seizures in those with a seizure disorder [45]. However, no effect on seizure activity was observed in a randomized, placebo-controlled trial in children by Wasdell et al. [28] or by Carr et al. [27]. Another study found a decrease in seizures in children who took exogenous melatonin to improve sleep [46]. In patients with epilepsy with idiopathic generalized tonic-clonic seizures, Maghbooli et al. found that the addition of melatonin to patients' valproic acid treatment was associated with a significant reduction in the severity of epilepsy and improvement in sleep quality [47].

### 9.6. Increase in Bone Fractures

There have been reports of increased risk of bone fracture associated with exogenous melatonin [45]. A study of adults in the United Kingdom by Frisher et al. [48] found that adults (average age 65) who received 3 or more melatonin prescriptions (2 mg, extended release) were 44% more likely to suffer a fracture than the control group. Until the association between melatonin and fractures has been clarified, the possibility of increased fracture risk is a reason to use melatonin with caution in older adults and those with previous low-impact fractures.

### 9.7. Impairment of Balance or Cognition

There have been concerns that exogenous melatonin could impair balance or cognition. Immediate-release melatonin is cleared quickly from the system. Oral, immediate-release melatonin has a half-life of approximately 45 min and time to peak blood concentration of approximately 30 min. A single oral dose of melatonin will increase blood concentrations for approximately 5 h. A study of men and women aged 60–71 in China found that, compared to placebo, a single 3 mg dose of melatonin significantly impaired balance 1 h after ingestion; however, it did not significantly affect cognitive function [49]. A long-term, placebo-controlled study in postmenopausal women found no increase in adverse events, daytime drowsiness, muscle weakness, or impaired balance the day after taking 1–3 mg of immediate-release melatonin; however, higher doses (over 3 mg) of controlled-release melatonin products can potentially cause next-day drowsiness. A study by Gooneratne et al. demonstrated high blood levels of melatonin for an average of 10 h following ingestion of a 4 mg product (3 mg controlled-release + 1 mg immediate-release) compared to 6.4 h following a 0.4 mg product (0.3 mg controlled-release + 0.1 mg immediate-release) [50]. For these reasons, caution has been advised when driving or operating heavy machinery within 6 h after taking melatonin, especially after taking controlled-release products.

*9.8. Worsening of Restless Leg Syndrome*

Taking melatonin may increase leg movements in restless leg syndrome (RLS). A small study of 8 subjects with severe RLS found significantly more leg movement when measured 1 h and 4.5 h after taking 3 mg of melatonin in the evening compared to no melatonin. However, the patients in the study did not report any increase in leg discomfort [51]. In contrast, preliminary evidence shows that melatonin may decrease periodic leg movement disorder symptoms involving involuntary leg movements while sleeping [52].

*9.9. Worsening of Asthma*

There has been concern that exogenous melatonin might worsen asthma. However, a small study among people with stable, mild, to moderate asthma found that taking 3 mg of melatonin at night for 1 month did not significantly affect lung function compared to placebo [53].

*9.10. Worsening of Depression and Bipolar Disorder*

Extremely high doses of melatonin (250 to 1200 mg daily) have been reported to worsen depression and bipolar disorder [54]. However, a study among people with depression using melatonin 0.5 to 6 mg daily for up to 3.5 years did not report a worsening of depression [55].

## 10. Summary

In summary, commonly reported side effects of long-term use of exogenous melatonin are minor, and data from the available studies regarding its long-term safety are generally reassuring. No clinically significant adverse effects have been consistently identified.

## 11. Conclusions

Exogenous melatonin is widely used worldwide in adults and children to treat insomnia, other sleep disorders, and various other medical conditions, including autistic spectrum disorder, minimal cognitive impairment, and Alzheimer's disease. It is classed as a dietary supplement in the U.S.; hence, it is available over the counter and considered a nutraceutical product by the government. Any regulatory agency does not closely monitor its production and sale as compared to substances described as drugs in the U.S., and the melatonin content of available preparations may vary considerably. The effectiveness of melatonin in initiating sleep is measurable but small in most people. It appears to have less of a role in increasing sleep duration than in sustained-release preparations. Optimal dosing has not been established; commonly utilized doses vary widely. Short-term side effects of melatonin appear mild, resolve when stopping the medication, and do not limit its use in most cases. Many studies of the long-term use of melatonin have shown no difference in the incidence of long-term side effects between exogenous melatonin and placebo. Melatonin taken in low to moderate doses (5 mg daily or less) appears safe for short- and long-term use. However, it is widely agreed that the long-term effects of taking exogenous melatonin have been insufficiently studied. The greater potential therapeutic values of melatonin usage have yet to be explored thoroughly and warrant further research. Some of the most promising potential roles for long-term higher dose melatonin include reduced Alzheimer's disease related symptoms and increased longevity. Though some experts have advocated for larger doses of melatonin—including over 20 mg per day— a lower dose is a more reasonable dose for adults at present, until additional studies are performed. Using either an immediate relief preparation and/or in combination with an extended-release formulation— e.g., 3 mg immediate release and 3 mg extended release—is a reasonable starting point. Further research is warranted, but preliminary data suggest that melatonin may greatly benefit aging individuals given its relatively safe profile, potential anti-aging effects, mitigation of cognitive decline, and unequivocal improvement in sleep quality.

**Author Contributions:** Study concept and design, D.G., A.G., P.M.L., D.M.W., S.A., S.S., A.N.E., B.K.D., C.J.B., E.M.C., A.M.K. and A.D.K.; Analysis and interpretation of data, D.G., A.G., P.M.L., D.M.W., S.A., S.S., A.N.E., B.K.D., C.J.B., E.M.C., A.M.K. and A.D.K.; Drafting of the manuscript, D.G., A.G., P.M.L., D.M.W., S.A., S.S., A.N.E., B.K.D., C.J.B., E.M.C., A.M.K. and A.D.K.; Critical revision of the manuscript for important intellectual content, D.G., A.G., P.M.L., D.M.W., S.A., S.S., A.N.E., B.K.D., C.J.B., E.M.C., A.M.K. and A.D.K.; Statistical analysis, D.G., A.G., P.M.L., D.M.W., S.A., S.S., A.N.E., B.K.D., C.J.B., E.M.C., A.M.K. and A.D.K. All authors listed have made a direct and intellectual contribution to the work and approved for publication. All authors have read and agreed to the published version of the manuscript.

**Funding:** This research received no external funding.

**Institutional Review Board Statement:** Not applicable.

**Informed Consent Statement:** Not applicable.

**Data Availability Statement:** Data sharing is not applicable to this article as no datasets were generated or analyzed during the current study.

**Conflicts of Interest:** The authors declare no conflict of interest.

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
