# Peer review of "Chronic Administration of Melatonin: Physiological and Clinical Considerations"

_2035-8377, doi:10.3390/neurolint15010031_

Round 1

Reviewer 1 Report

This is a narrative review of clinical studies of primarily chronic usage of melatonin, with an emphasis on publications of the last 5 years. Melatonin's effects on a variety of disease conditions, its side effects and its drug interactions are covered and conclusions summarized. The review is well organized and presented, and seems thorough and accurate. Thus it can be recommended for publication.

The following minor points should be corrected:
(Line 163) It is stated that melatonin may increase the risk of bleeding for patients taking warfarin. However, there seems to be no reference to support this statement, as reference 28 appears not to mention warfarin. A reference supporting this statement should be cited, or the statement should be deleted.

(Line 199) Fluorodeoxyglucose is a synthetic reagent used for assessing energy use in the brain. Thus to state "diagnosis...for presence of...fluorodeoxyglucose" is misleading and needs to be restated to better describe the role of this reagent, or its mention deleted.

Author Response

Date: March 9, 2023

Manuscript number: neurolint-2250396 

Title: Chronic Administration of Melatonin: Physiological and ClinicalConsiderations

Dear Editor in Chief,

Thank you for your recent correspondence regarding our paper on Chronic Administration of Melatonin: Physiological and Clinical Considerations, submitted to Neurology International. Here is a corrected manuscript with a point-by-point response to comments from the Reviewers. The comments by the reviewers were of great value and each point had been addressed below.

Specifically, with regard to Reviewers' Comments to Author:

Reviewer 1 comments:

This is a narrative review of clinical studies of primarily chronic usage of melatonin, with an emphasis on publications of the last 5 years. Melatonin's effects on a variety of disease conditions, its side effects and its drug interactions are covered and conclusions summarized. The review is well organized and presented, and seems thorough and accurate. Thus it can be recommended for publication.

The following minor points should be corrected:
(Line 163) It is stated that melatonin may increase the risk of bleeding for patients taking warfarin. However, there seems to be no reference to support this statement, as reference 28 appears not to mention warfarin. A reference supporting this statement should be cited, or the statement should be deleted.

Thank you for the comment, we have reworked this section and added supporting sources on the potential interactions between warfarin and melatonin.

(Line 199) Fluorodeoxyglucose is a synthetic reagent used for assessing energy use in the brain. Thus to state "diagnosis...for presence of...fluorodeoxyglucose" is misleading and needs to be restated to better describe the role of this reagent, or its mention deleted.

We have reworked this sentence to include fluorodeoxyglucose’s usage as a reagent in PET scans, rather than something scanned for progression of AD.

Reviewer 2 comments:

This review on melatonin drug effects, and potential side effects and safety issues, covers a wide spectrum and was well written. However, the authors should revise the manuscript in consideration of the following concerns:

  1. A number of statements  are not referenced, eg., lines 74-78, 87-105, 138-139... plus some other lines the authors better check out.

Thank you for the comment. We have gone through the above-mentioned lines and added sources supporting their statements.

  1. 198, melatonin as AD biomarker is questionable, since evidence is thin.

We have reworded this section as to describe the role of melatonin as a potential biomarker for AD rather than a tried and true one. We have also added in a statement that more studies should be performed to evaluate circulating melatonin for this usage.

  1. On the section Potential Drug Interaction, not much information is provided, and authors should either remove this section or enrich it.

We have removed this section as it is not substantial in length nor weight in the discussion.

  1. Authors may consider mentioning briefly pharmacokinetics of melatonin.

We have added in a brief section on the pharmacokinetics of melatonin, including an interesting revelation on its metabolism by CYP1B1 that suggests melatonin may be metabolized in extrahepatic tissues.

Sincerely,

Sahar Shekoohi, PhD

Post-Doctoral Fellow

Louisiana State University Health Sciences Center at Shreveport, Department of Anesthesiology, Shreveport, LA, 71103, USA

Sahar.Shekoohi@Lsuhs.edu

Reviewer 2 Report

This review on melatonin drug effects, and potential side effects and safety issues, covers a wide spectrum and was well written. However, the authors should revise the manuscript in consideration of the following concerns:

1. A number of statements  are not referenced, eg., lines 74-78, 87-105, 138-139... plus some other lines the authors better check out.

2. p.198, melatonin as AD biomarker is questionable, since evidence is thin.

3. On the section Potential Drug Interaction, not much information is provided, and authors should either remove this section or enrich it.

4. Authors may consider mentioning briefly pharmacokinetics of melatonin. 

Author Response

(The authors gave the same response as above.)
